# Centroids Matching: an efficient Continual Learning approach operating in the embedding space

**Jary Pomponi**                                    *jary.pomponi@uniroma1.it*
*Department of Information Engineering,*
*Sapienza University of Rome, Italy*

**Simone Scardapane**                               *simone.scardapane@uniroma1.it*
*Department of Information Engineering,*
*Sapienza University of Rome, Italy*

**Aurelio Uncini**                                  *aurelio.uncini@uniroma1.it*
*Department of Information Engineering,*
*Sapienza University of Rome, Italy*

**Reviewed on OpenReview:** *https://openreview.net/forum?id=7gzQltQSwr*

## Abstract

Catastrophic forgetting (CF) occurs when a neural network loses the information previously learned while training on a set of samples from a different distribution, i.e., a new task. Existing approaches have achieved remarkable results in mitigating CF, especially in a scenario called task incremental learning. However, this scenario is unrealistic, and limited work has been done to achieve good results in more realistic scenarios. In this paper, we propose a novel regularization method called Centroids Matching, that, inspired by meta-learning approaches, fights CF by operating in the feature space produced by the neural network, achieving good results while requiring a small memory footprint. Specifically, the approach classifies the samples directly using the feature vectors produced by the neural network, by matching those vectors with the centroids representing the classes from the current task, or all the tasks up to that point. Centroids Matching is faster than competing baselines, and it can be exploited to efficiently mitigate CF, by preserving the distances between the embedding space produced by the model when past tasks were over, and the one currently produced, leading to a method that achieves high accuracy on all the tasks, without using an external memory when operating on easy scenarios, or using a small one for more realistic ones. Extensive experiments demonstrate that Centroids Matching achieves accuracy gains on multiple datasets and scenarios.

## 1 Introduction

An agent which operates in the real world must be able to continuously learn from the environment. Learning from a stream of samples, usually in the form of static datasets, also called tasks, is referred to as Lifelong Learning or Continual Learning (CL). A continual learning scenario comes often with a phenomenon called Catastrophic Forgetting (CF) (McCloskey and Cohen, 1989), that arises when an agent loses the knowledge learned from past samples while extracting information from newer ones. This phenomenon inhibits the correct working of agents that operate in such scenarios, but it can be mitigated, or removed, using methods built for that purpose. A key point is that such methods must present a contained memory footprint, because we can't save all past samples encountered during the training Parisi et al. (2019), and the agents cannot grow indefinitely, consuming all the memory. Thus, the external memory, intended as the collection of all the things saved on the hardware that must be preserved across all the tasks, must be contained.

Over the last years, a large amount of research has been done about methods to alleviate the CF phenomenon. Usually, CL techniques are grouped into three categories (regularization-based methods, rehearsal methods, and architectural methods), and a method can belong to one, or more, categories at the same time (Parisi et al., 2019). The methods in the first set are designed in such a way that important parameters of the model, with respect to past tasks, are preserved during the training of newer tasks using any sort of regularization technique, by directly operating over the parameters of the model, or by adding a regularization term to the training loss (Li and Hoiem, 2017; Kirkpatrick et al., 2017; Zenke et al., 2017; Serra et al., 2018; Saha et al., 2020; Chaudhry et al., 2021). Rehearsal-based methods save portions of past tasks and use the information contained in the memory to mitigate CF while training on new tasks (Rebuffi et al., 2017; Chaudhry et al., 2019; van de Ven et al., 2020; Chaudhry et al., 2021; Rosasco et al., 2021); the samples associated to past tasks can also be generated using a generative model, and in that case the methods are called pseudo-rehearsal. Finally, architectural-based methods freeze important parameters or dynamically adjust and expand the neural network's structure, to preserve the knowledge associated to past tasks, while the model learns how to solve the current task (Rusu et al., 2016; Aljundi et al., 2017; Yoon et al., 2017; Veniat et al., 2020; Pomponi et al., 2021).

Aside from developing techniques to solve CF, another issue is formalizing scenarios describing how tasks are created, how they arrive, and what information is provided to the model itself (e.g., the task identifier). Usually, a method is designed to solve a subset of all possible CL scenarios (Van de Ven and Tolias, 2019). In this paper, we operate on scenarios in which the identity of a task is always known during the training, but it may not be known during inference phase, and the classes are disjoint (the same class appears in only one task). If we can use the task identity during the inference phase, we have a scenario called *task incremental*, otherwise, the scenario is called *class incremental*; the latter is much harder to deal with, being closer to a real-world scenario Van de Ven and Tolias (2019). The task incremental scenarios have been studied exhaustively, due to the simplicity of the problem, while fewer methods have been proposed to efficiently solve the latter. In this paper, we propose a method that achieves better results on both task and class incremental scenarios.

Our proposal, Centroids Matching, is a pure regularization-based method when it comes to fight CF in task incremental problems (which are easier), while it uses an external memory, containing samples from past tasks, when we require to solve class incremental scenarios; when fighting the CF phenomenon in a task incremental scenario, our approach requires no memory, since we force the model to keep its extraction ability by using the current samples and the current model, without the need to store the past model when the learning of a task is over, nor an external memory containing past samples. Our approach differs from the existing literature because it does not train the neural network using a standard training procedure, in which a cross-entropy error is minimized, but it operates directly on the embeddings vectors outputted by the model, by producing an embedding for each point. These vectors are moved to match the centroids of the associated classes, which are calculated using a sub-set of training samples (support set). When fighting the CF phenomenon in a task incremental scenario, our approach requires no memory, since we force the model to keep its extraction ability by using the current samples and the current model, without the need to store the past model when the learning of a task is over, nor an external memory containing past samples.

## 2 Related Works

An agent that has the Continual Learning (CL) property is capable of learning from a sequence of tasks (De lange et al., 2021) without forgetting past learned knowledge. When past learned knowledge is lost, and with it also the ability to solve tasks already solved in the past, we have a phenomenon called Catastrophic Forgetting (CF) (French, 1999; McCloskey and Cohen, 1989), which occurs because the information saved in the parameters of the model is overwritten when learning how to solve new tasks, leading to a partial or total forgetting of the information. A CL method should be able of alleviating, or removing, CF while efficiently learning how to solve current tasks. Initial CL works focused on fighting the CF phenomenon on the easiest supervised CL scenario, called task incremental learning, but, recently, we have seen a shift toward class incremental scenario, being closer, and more suitable, to real-world applications; nevertheless, a limited number of proposed approaches focus on that specific scenario (Masana et al., 2020; Belouadah et al., 2021). The main difference between the two is that in the first scenario we can classify only samples in

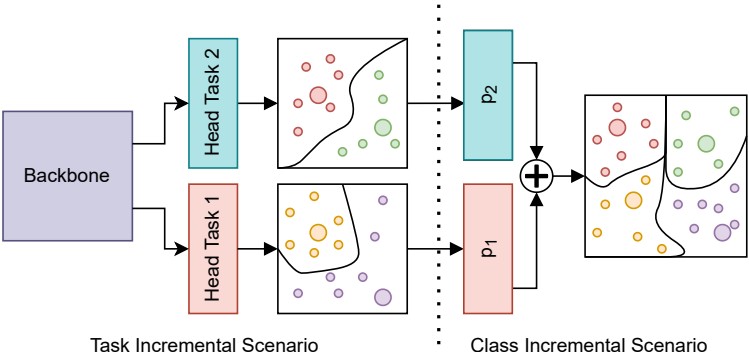

Figure 1: The proposed approach applied on both Task (left) and Class (right) Incremental learning scenarios. The bigger circles are the centroids of the classes, while the smaller ones are samples from the same class of the corresponding centroid. We see that a **CIL** scenarios is solved by merging the embedding spaces together, into a new space that contains all the classes so far; the merging process is explained in Section 4.2.

the context of a task, thus the task identity must be known a priori, while in the latter one the model must discriminate at inference time between all classes seen during the training procedure, without having the task identity. Depending on how a CL method achieves this goal, we can group, following Parisi et al. (2019), the CL methods into three categories: regularization methods, rehearsal methods, and architectural methods. Our approach belongs to the first set when we are dealing with task incremental scenarios, and both the first and the second one when the scenario is a class incremental one.

Our approach regularizes the model by constraining the embeddings, but other approaches, based on the same principle, have been proposed over the years. One of the first was proposed in Hou et al. (2019), and it does not work directly on the embeddings of the model, but on the logits produced by it, and uses them to regularize the training by reducing the distance between the current model and the past one, while also correcting biases that arise when training a model to solve a CL scenario (e.g. class imbalances). In Pomponi et al. (2020), the authors proposed a regularization-rehearsal approach that works directly on the embeddings space produced by the model. Given a sample and the task from which the sample comes, the proposed method uses a regularization term that forces the model to reduce the distance between the current embeddings vector and the one obtained when the training on the source task was over; moreover, the approach requires a very small memory to work, but it can only regularize models that operate on task incremental scenarios because it requires tasks spaces to be separated. In Han and Guo (2021) a regularization-rehearsal method which uses the embeddings vectors to regularize the model is proposed. The vectors are used to calculate multiple contrastive losses used as regularization factors; also, a mechanism that overwrites a portion of the embeddings is used, enabling selective forgetting; unfortunately, the approach requires a big external memory in order to achieve competitive results.

More CL scenarios exist, such as a stream-supervised scenario, often called Online Incremental Learning, in which the model sees a sample only once, and the idea of using the embeddings to regularise the model has also been exploited in these scenarios. Starting from the same ground idea which inspired our approach, in De Lange and Tuytelaars (2021), the authors proposed a CL approach that operates over a stream of samples, by continuously updating the prototypes extracted using the same stream, by using a novel loss which synchronizes the latent space with the continually evolving prototypes. When a batch is retrieved, it is used to update the model using the proposed loss function and the centroids, using a combination of the current centroids and the embedding vectors associated with each image in the batch. When this process is over, the samples in the batch are used to populate a class-wise memory; the base idea, classifying using prototypes, is similar to our approach, with the difference that the authors used a different loss function that works both on positive and negative samples, and the prototypes are calculated as mobile average while samples are retrieved from the stream, and, in the end, the concept of task is missing in their proposal. Similarly, the

authors of Taufique et al. (2022) proposed an approach that works in the context of unsupervised domain adaptation, in which a buffer containing prototypes is used to calculate a contrastive loss against the current batch of samples. In the approach proposed in Kurniawan et al. (2021), which aims to solve online continual learning scenarios, the authors used many loss functions to train the model, and one of them is based on the similarity calculated in the embedding space, which pulls closer the samples belonging to the same class.

Other CL methods, that do not use the embeddings to regularize the training, have been proposed over the years. The approaches belonging to the regularization-based set fight CF by forcing the model's parameters, or any output of the agent, which are relevant for past tasks, to be as close as possible to the optimal parameters obtained when these past tasks were over. One of the first methods that use a regularization approach to fight CF is Elastic Weight Consolidation (EWC), proposed in Kirkpatrick et al. (2017), that assigns an importance scalar to each parameter of the model, slowing the change of the parameters that are considered more important to preserve the ability to solve past tasks; in some cases this constraining approach could be too strong, leading to an incapacity of the model to learn how to solve new tasks. Other methods, such as the ones proposed in Saha et al. (2020) and Farajtabar et al. (2020), regularize the model by forcing the gradients to go toward a space of the parameters where the CF is minimized for all the tasks, while the current one is being solved as efficiently as possible, by moving the weights in the space that satisfies all the constraints. Memory-based methods save a small number of samples from each solved task, or generate synthetic samples using generative models, to be used jointly with the current training samples, in order to preserve past learned knowledge. These methods are often called rehearsal methods, or pseudo-rehearsal when a generative model is involved. The most representative method in this set, which is a hybrid rehearsal-regularization approach, is proposed in Lopez-Paz and Ranzato (2017), which uses the samples from the external memory to estimate the gradients associated to past tasks, which are used to modify the gradients associated with the current training samples, to solve, jointly, the current and the past tasks; moreover, this was one of the methods, along with plain rehearsal approach, that can improve the scores obtained on past tasks, supposing that the memory dimension is big enough to be fully representative of past tasks. A more straightforward approach, yet very effective, is to use the external memory to augment the current batch by concatenating a random batch extracted from the memory and the current batch, as proposed, for instance, in Chaudhry et al. (2019); Riemer et al. (2018); Yoon et al. (2021). Being a straightforward approach, many other similar approaches, as well as theoretical studies to understand what CF really is and how to fight it, have been proposed over the years (Rebuffi et al., 2017; Rolnick et al., 2019; Ostapenko et al., 2022).

## 3  Continual Learning Setup

We define a supervised CL scenario as a set of N tasks $\text{T} = \{\mathcal{T}_i\}_{i=1\ldots\text{N}}$, in which a task can be retrieved only when the training on the current one is over; when a new task is collected, the past ones cannot be retrieved anymore (except for testing purpose). Each task $\mathcal{T}_i$ is a set of tuples $(\mathbf{x}, y)$, where $\mathbf{x}$ is a sample, and $y \in Y_i$ is the label associated to it, where $Y_i$ is the set of classes contained in the task $\mathcal{T}_i$. Also, the tasks are disjoint, meaning that: $\bigcap_{i=1\ldots\text{N}} Y_i = \varnothing$ (a class cannot belong to two different tasks). The goal of a CL method is to help the model to generally perform well on all learned tasks so far, by adapting to new tasks while preserving the previously learned knowledge. A method that solves a task at the expense of another is not desirable, and thus a trade-off must be achieved, by looking at the overall performance.

Assuming that the tasks' boundaries are always known and well defined during the whole training procedure, i.e., we always know when a task is over or retrieved, we follow Van de Ven and Tolias (2019) to define two different scenarios, based on how the inference procedure is carried out:

- Task Incremental Learning (**TIL**): in which the task's identity of a sample is given during the inference phase.

- Class Incremental Learning (**CIL**): in which the task's identity of a sample is not given during the inference phase.

The difference is minimal, but yet crucial. In fact, we can consider the first one as a more simple and theoretical scenario, which is also the most studied one. Its main limitation is that, to correctly classify

a sample, we must know the task from which the sample comes, and usually, this is not the case. In fact, such scenarios are more suitable to develop and test novel methods, before adapting them to an agent that operates on more realistic scenarios. The second scenario is more difficult and the agents suffer CF drastically, because not only the model must be regularized, but the space of the prediction must be extended to include also the upcoming classes, leading to a faster forgetting of past tasks.

It must be noted that the scenario definition is untied from the architecture of the neural network involved, which can have any topology, as long as the scenario's rules are followed. Nevertheless, usually, a multi-head strategy is adopted to operate in **TIL** scenarios, in which a backbone is shared, and, for each task, a smaller neural network, usually called head, is used to classify the samples belonging to that task; each head takes as input the output of the backbone, making the prediction spaces of the tasks well separated. The shared backbone is also used when operating in **CIL** scenarios, but the classification head is usually just one, whose output neurons are expanded (the newer classes are added during the training) when a new task is collected.

# 4 Centroids Matching (CM) framework

Our approach is inspired by the Prototypical Networks proposed in Snell et al. (2017), following the idea that there exists an embedding space, also called feature space, in which vectors cluster around the most probable centroid, representing a class, and called *prototype*. Following the approach used in Prototypical Networks, our model does not follow a standard classification approach in which a cross-entropy loss is minimized, but it uses the model to extract a features vector from an input sample, and then it forces the vector to be as close as possible to the correct centroid, representing the class in the embedding space of the task.

In the following section, we explain how this approach can be used to easily mitigate CF in multiple CL scenarios.

## 4.1 TIL scenario

Following Section 3, suppose the model consists of two separate components: a feature extractor, also called backbone, $\psi : \mathbb{R}^I \to \mathbb{R}^D$, where I is the size of an input sample and D is the dimension of the features vector produced by the backbone, and a classifier head $f_i : \mathbb{R}^D \to \mathbb{R}^E$, one for each task, where E is the dimension of the task specific feature vector. The backbone operates as a generic feature extractor, while each head is a specific model that, given the generic vector of features extracted by the backbone, transforms the vector into a vector of features for that specific task. Given an input sample $\mathbf{x}$ and the task $i$ from which the sample comes, the final vector, used for training and testing, is carried out by combining the functions: $e_i(\mathbf{x}) = f_i \circ \psi(\mathbf{x})$. The backbone remains unique during the whole training, while a new head $f$ is added each time a new task is encountered. Both the backbone and all the heads are updated during the whole training process.

When a new task $\mathcal{T}_i$ is available, we extract and remove a subset of the training set, named support set and identified as $S_i$, containing labelled training samples that won't be used to train the model. This support set is used to calculate the centroids, one for each class in the task. A centroids, for a given class $k$, in the space of the task $i$, is the average of the feature vectors extracted from the samples in the support set, calculated using the corresponding head:

$$\mathbf{c}_i^k = \frac{1}{|S_i^k|} \sum_{(\mathbf{x},y) \in S_i^k} e_i(\mathbf{x}) \tag{1}$$

where $S_i^k$ is the subset of $S_i$ that contains only samples having label $k$. During the training, these centroids are calculated at each iteration, in order to keep them up to date. Then, given the Euclidean distance function $d : \mathbb{R}^M \times \mathbb{R}^M \to \mathbb{R}_+$, and a sample $\mathbf{x}$, our approach produces a distribution over the classes based on the softmax distance between the features produced using $e_i(\mathbf{x})$ and the centroids associated to the current task:

$$p(y = k|\mathbf{x}, i) = \frac{\exp(-d(\mathbf{c}_i^k, e_i(\mathbf{x})))}{\sum_{k' \in Y_i} \exp(-d(\mathbf{c}_i^{k'}, e_i(\mathbf{x})))} \tag{2}$$

We note that, in this scenario, it makes no sense to calculate the distances between different tasks' heads, since each head produces centroids placed in their own embedding space (see the left side of Fig. 1), without interfering with the others. The loss associated to a sample is then the logarithm of the aforementioned probability function:

$$L(\mathbf{x}, k, i) = -\log p(y = k|\mathbf{x}, i) \tag{3}$$

If the current task is not the first one, in order to preserve past learned knowledge, we need to regularize the model. When a new task is collected, we clone the current model, both the backbone and all the heads created so far, which we indicate as $\overline{e_i}(\cdot)$, for each task $i$. Then, while training on the current task, we augment the loss using the distance between the features extracted using the cloned model and the one extracted by the current one, both calculated using the current set of training samples, without the support of an external memory containing past samples. The regularization term is the following:

$$R(\mathbf{x}, t) = \frac{1}{t} \sum_{i < t} d\left(\overline{e_i}(\mathbf{x}), e_i(\mathbf{x})\right) \tag{4}$$

The proposed regularization term works because it relies on the fact that a Prototypical Network can be reinterpreted as a linear model Mensink et al. (2013). Using this simple regularization term, we force the model to preserve the ability to extract the same information that it was able to extract when the previous task was over. Moreover, since the regularization term is calculated only using samples from the current task, no external memory is needed to regularize the model. The overall regularization approach works because the past heads are trained at the same time as the new ones, while leaving the weights of the model unconstrained, as long as the output distance is minimized. Then, the final loss for a task which is not the first one is:

$$L_{ti}(\mathbf{x}, k, t) = -\log p(y = k|\mathbf{x}, t) + \lambda R(\mathbf{x}, t) \tag{5}$$

where $\lambda$ is a scalar used to balance the two terms. When a task is over, the final centroids for the classes in the task are calculated and saved. Thus, the external memory contains, when all the tasks are over, only the centroids for the classes seen during the training, and thus the required memory is negligible.

To classify a sample $\mathbf{x}$ from a task $t$, we use the same approach used during the training process, based on the distance between the centroids of that task and the features extracted from the sample $\mathbf{x}$:

$$y = \underset{k \in Y_i}{\operatorname{argmax}} \ p(y = k|\mathbf{x}, t) \tag{6}$$

This is possible because we always know from which task the sample comes. In the next section, we will show how this can be achieved when the task identity is not known.

### 4.2 CIL scenario

The class incremental scenario is a more difficult if compared to the **TIL** scenario, because the identity of the task is available only during the training process but not during the inference phase, requiring a different

approach. Most of the methods fails to overcome CF in this scenario, mainly because a single head classifier is used to classify all the classes, leading to faster CF, because the capacity of a single head is limited. Instead, in our approach, we keep the heads separated as in the **TIL** scenario, and we train by projecting the embeddings vectors into a shared embedding space, containing all the classes so far, leading to an easier classification phase.

As before, we train on the first task $t$ using the loss 5. If the task is not the first one, we also add a projection loss to the training loss, which is used to project all the embedding spaces into a single one. First of all, we use an external memory $\mathcal{M}$, which can contain a fixed number of samples for each task, or a fixed number of samples during the whole training, removing a portion of past images when new ones need to be stored. In our experiment, we use a fixed sized memory, which is resized each time a new task must be saved in the memory.

If the current task $i$ is not the first, we augment the current dataset with the samples from the memory. Since the memory is smaller than the training set, we use an oversampling technique when sampling samples from the memory, in a way that a batch always contains samples from past tasks. We define the projected loss, which is a modified version of the equation 2, as:

$$\overline{p}(y = k | \mathbf{x}, i) = \frac{\exp(-d(p_i(\mathbf{c}_i^k), \frac{1}{i} \sum_{j \leq i} p_i(\mathbf{x})))}{\sum_{k' \in Y_i} \exp(-d(p_i(\mathbf{c}_i^{k'}), \frac{1}{i} \sum_{j \leq i} p_i(\mathbf{x})))} \tag{7}$$

where $Y_i = \bigcup_{j=1,\dots,i} Y_j$ contains all the classes up to the current task, and the function $p_i(\cdot)$ is a projection function, one for each task, that projects the embeddings, and the centroids, from the task wise embedding space to the shared embedding space. For this reason, the labels $y$ must be scaled accordingly using a simple scalar offset. For a generic task $i$, the projecting function $p_i$ is defined as:

$$p_i(x) = e_i(x) \cdot \text{Sigmoid}(s_i(e_i(x))) + t_i(e_i(x)) \tag{8}$$

in which the functions $s_i, t_i : \mathbb{R}^E \to \mathbb{R}^E$ are, respectively, the scaling and the translating function, implemented using two small neural networks, trained along with the backbone and the heads. The final loss for this scenario is defined as:

$$L_{ci}(\mathbf{x}, k, t) = -\log \overline{p}(y = k | \mathbf{x}, t) \tag{9}$$

At inference time, if we know the task associated to a sample, we can perform the inference step as in the **TIL** scenario, otherwise, we classify directly the samples, without inferring the corresponding task:

$$y = \underset{k \in Y}{\operatorname{argmax}} \ \overline{p}(y = k | \mathbf{x}, \mathrm{N}) \tag{10}$$

where $Y$ is the set of all the classes seen during the training, and N is the total number of tasks. In this way, all the tasks are projected into the same embedding space, thus the classification of a sample does not require the task identity.

## 5 Experiments

### 5.1 Experimental setting

**Dataset**: We conduct extensive experiments on multiple established benchmarks in the continual learning literature, by exploring both **TIL** as well as the harder **CIL**. The datasets we use to create the scenarios are: CIFAR10, CIFAR100 (Krizhevsky, 2009), and TinyImageNet (a subset of ImageNet (Deng et al., 2009) that contains 200 classes and smaller images). To create a scenario we follow the established approach, where the classes from a dataset are grouped into N disjoint sets, with N the number of tasks to be created. We use CIFAR10 to build a scenario composed of 5 tasks, each one with 2 classes; using CIFAR100 we create 10 tasks with 10 classes in each one; in the end, the classes in TinyImageNet are grouped into 10 tasks, each one containing 20 classes.

**Baselines**: we test our approach against many continual learning methods: Gradient Episodic Memory (GEM) (Lopez-Paz and Ranzato, 2017), Elastic Weight Consolidation (EWC) (Kirkpatrick et al., 2017), Online EWC (OEWC) (Schwarz et al., 2018), Experience Replay (ER) Chaudhry et al. (2019), SS-IL Ahn et al. (2021), CoPE De Lange and Tuytelaars (2021), despite being introduced for solving stream domain scenarios, under both multi epochs approach (ME) and stream approach (S), and, in the end, Embedding Regularization (EmR) (Pomponi et al., 2020); regarding the latter, it only works on **TIL** scenarios, and the other results are omitted. We also use two baseline approaches: Naive Training, in which the model is trained without fighting the CF, and Cumulative Training, in which the current training task is created by merging all past tasks as well as the current one.

**Hyper-parameters**: for each method, we searched for the best hyper-parameters, following the results presented in respective papers. For EWC, we used 100 as regularization strength weight for all the scenarios. For GEM we used a memory for each task, composed of 500 samples for CIFAR10 and 1000 for the other experiments. In the EmR memory, we saved 200 samples from each task, while for CoPE we have a memory for each class, and each one can contain up to 300 samples. Lastly, for ER and SS-IL, we used a fixed memory size of 500 for CIFAR100 scenarios and 1000 otherwise. Regarding our approach, the support set contains 100 images from the training set of each task, and we set the penalty weight $\lambda$ to 0.1 for CIFAR10, 0.75 for CIFAR100 and TinyImageNet; regarding the **CIL** scenarios, we used a fixed size memory of 500 for each scenario. For both ER and CM, the memory is shared across all the tasks, and it is resized, by discarding images from past tasks, each time a new task must be saved.

**Models and Training**: for each dataset we use ResNet20 (He et al., 2016) architecture, trained using SGD with learning rate set to 0.01 and momentum to 0.9. For CIFAR10-100, we trained the model for 10 epochs on each task, while for TinyImagenet we used 30 epochs; the only exception is SSIL, for which we used 100 epochs for each scenario. For each classification head, we used two linear layers, with the first layer that takes as input the output of the backbone, followed by a ReLU activation function, and then an output layer whose output size depends on the number of classes in the current task. Regarding our proposal, each head is composed of two linear layers, and it projects the output of the backbone into a vector of 128 features (the output of the ResNet model has 64 values). We repeat each experiment 5 times; each time the seed of the experiment is changed in an incremental way (starting from 0). Regarding our proposal, since it is not a rehearsal method when operating in the **TIL** scenario, after each task we save the statistics of the batch norm layers, which are retrieved when a sample from the corresponding task must be classified. Also, we used the following augmentation schema for the proposed datasets: the images are standardized, randomly flipped with probability 50%, and then a random portion of the image is cropped and resized to match the original size.

A scenario, usually, is built by grouping the classes in an incremental way (the first n classes will form the first task, and so on). We use this approach for the first experiment, instead, when the experiment is not the first, each of the N tasks is created using a randomly selected subset of classes. Using this approach, a different scenario is built for each experiment, and more challenging scenarios could be created since the correlation between the classes disappears.

**Metrics**: to evaluate the efficiency of a CL method, we use two different metrics from Díaz-Rodríguez et al. (2018), both calculated on the results obtained on the test set of each task. The first one, called Accuracy, shows the final accuracy obtained across all the test splits of the tasks, while the second one, called Backward Transfer (BWT), measures how much of that past accuracy is lost during the training on upcoming tasks. To calculate the metrics we use a matrix $\mathbf{R} \in \mathbb{R}^{N \times N}$, in which an entry $\mathbf{R}_{i,j}$ is the test accuracy obtained on the test split of the task $j$ when the training on the task $i$ is over. Using the matrix $\mathbf{R}$ we calculate the metrics as:

$$\text{Accuracy} = \frac{1}{N} \sum_{j=1}^{N} \mathbf{R}_{N,j} \,, \qquad\qquad \text{BWT} = \frac{\sum_{i=2}^{N} \sum_{j=1}^{i-1} (\mathbf{R}_{i,j} - \mathbf{R}_{j,j})}{\frac{1}{2} N(N-1)} \,.$$

Both metrics are important to evaluate a CL method, since a low BWT does not imply that the model performs well, especially if we also have a low Accuracy score because, in that case, it means that the

Table 1: Mean and standard deviation (in percentage), calculated over 5 experiments, of achieved Accuracy and BWT for each combination of scenario and method; some results are missing because the corresponding method does not work on that specific scenario. The best results for each combination of dataset-scenario are highlighted in bold.

| Method | TIL | | | | | | CIL | | | | | |
| | CIFAR10 | | CIFAR100 | | TinyImageNet | | CIFAR10 | | CIFAR100 | | TinyImageNet | |
| | BWT | Accuracy | BWT | Accuracy | BWT | Accuracy | BWT | Accuracy | BWT | Accuracy | BWT | Accuracy |
|---|---|---|---|---|---|---|---|---|---|---|---|---|
| Naive | $-34.60_{\pm7.39}$ | $67.00_{\pm4.98}$ | $-59.63_{\pm14.50}$ | $26.75_{\pm13.20}$ | $-61.39_{\pm2.15}$ | $21.84_{\pm1.12}$ | $-95.96_{\pm0.97}$ | $18.00_{\pm0.77}$ | $-79.64_{\pm1.11}$ | $8.34_{\pm0.09}$ | $-60.57_{0.7}$ | $6.26_{\pm0.04}$ |
| Cumulative | $-2.75_{\pm0.99}$ | $93.83_{\pm1.12}$ | $2.03_{\pm3.84}$ | $75.05_{\pm10.71}$ | $6.33_{\pm0.77}$ | $63.03_{\pm1.06}$ | $-2.84_{\pm1.59}$ | $86.42_{\pm0.32}$ | $-3.28_{\pm0.04}$ | $59.86_{\pm0.45}$ | $-5.88_{\pm0.11}$ | $29.22_{\pm0.64}$ |
| EWC | $-16.15_{\pm7.11}$ | $77.06_{\pm4.47}$ | $-5.11_{\pm0.91}$ | $58.62_{\pm0.91}$ | $-5.68_{\pm3.56}$ | $27.41_{\pm2.1}$ | $-92.52_{\pm2.58}$ | $17.07_{\pm0.89}$ | $-63.54_{\pm1.36}$ | $6.13_{\pm0.32}$ | $-43.58_{\pm6.70}$ | $0.5$ |
| OWC | $-15.67_{\pm9.70}$ | $76.07_{\pm6.60}$ | $-6.37_{\pm2.69}$ | $59.56_{\pm1.61}$ | $-7.60_{\pm5.31}$ | $24.37_{\pm19.37}$ | $-90.01_{\pm3.12}$ | $15.76_{\pm1.12}$ | $-61.43_{\pm2.03}$ | $5.97_{\pm0.94}$ | $-46.23_{\pm6.48}$ | $0.5$ |
| ER | $-2.95_{\pm0.67}$ | $90.56_{\pm0.64}$ | $-8.42_{\pm0.08}$ | $70.55_{\pm0.79}$ | $-17.15_{\pm0.05}$ | $43.31_{\pm0.72}$ | $-50.14_{\pm1.81}$ | $52.60_{\pm1.38}$ | $-67.22_{\pm0.39}$ | $25.09_{\pm0.22}$ | $-53.39_{\pm0.72}$ | $8.08_{\pm0.28}$ |
| GEM | $-4.87_{\pm1.56}$ | $90.15_{\pm1.19}$ | $-10.58_{\pm0.35}$ | $71.85_{\pm0.37}$ | $-57.39_{\pm0.59}$ | $14.08_{\pm0.15}$ | $-80.17_{\pm1.59}$ | $22.86_{\pm1.41}$ | $-62.71_{\pm1.56}$ | $17.09_{\pm0.92}$ | $-53.14_{\pm0.85}$ | $5.55_{\pm0.21}$ |
| EmR | $-2.30_{\pm0.98}$ | $91.39_{\pm1.51}$ | $-2.75_{\pm0.32}$ | $72.03_{\pm0.95}$ | $-8.43_{\pm1.00}$ | $46.88_{\pm2.03}$ | $-$ | $-$ | - | - | $-$ | $-$ |
| SS-IL | $-$ | $-$ | $-$ | $-$ | $-$ | $-$ | $-32.89_{\pm2.36}$ | $37.05_{\pm1.73}$ | $-30.01_{\pm2.04}$ | $24.13_{\pm1.49}$ | $-22.12_{\pm1.5}$ | $9.59_{\pm0.98}$ |
| CoPE (S) | $-$ | $-$ | $-$ | $-$ | $-$ | $-$ | $-$ | $49.14_{\pm3.70}$ | $-$ | $5.25_{\pm1.40}$ | $-$ | $1.43_{\pm0.02}$ |
| CoPE (ME) | $-$ | $-$ | $-$ | $-$ | $-$ | $-$ | $-$ | $36.12_{\pm1.12}$ | $-$ | $4.31_{\pm1.02}$ | $-$ | $1.34_{\pm0.12}$ |
| CM | $-2.09_{\pm0.71}$ | $\mathbf{92.72_{\pm1.33}}$ | $-5.88_{\pm0.90}$ | $\mathbf{74.76_{\pm1.17}}$ | $-13.45_{\pm3.62}$ | $\mathbf{47.80_{\pm2.93}}$ | $-18.71_{\pm10.84}$ | $\mathbf{64.64_{\pm12.78}}$ | $-62_{\pm1.23}$ | $\mathbf{27.91_{\pm0.39}}$ | $-52.13_{\pm0.91}$ | $\mathbf{12.04_{\pm0.32}}$ |

approach regularizes too much the training approach, leaving no space for learning new tasks. In the end, the combination of the metrics is what we need to evaluate. In addition to these metrics, we also take into account the memory used by each method. The memory is calculated as the number of additional scalars, without counting the ones used to store the neural network, that must be kept in memory after the training of a task while waiting for the new one to be collected; the memory used during the training process is not counted as additional, since it can be discarded once the training is over.

The formulas used to calculate the approximated required memory are:

- EWC: this approach saves a snapshot of the model after each task. The required memory is: N × P.

- OEWC: it is similar to the EWC, but it saves only one set of parameters, which is updated after the training on each task. The final memory is P.

- Rehearsal approaches (**CIL**): these methods need an external memory in which a subset from each task is saved, and then used to regularize the training. The required memory depends on the number of images saved, and it is calculated as I×M×N.

- EmR: this approach requires not only the images but also the features vector associated with each image (the output of the backbone). The required memory size is: (D + I)×M×N.

- CM (**TIL**): requires only to save, after each task, the centroids of the classes in the tasks; thus, the memory size is E × T.

where N is the number of tasks, P is the number of parameters in the neural network, I is the dimension of the input images, D is the dimension of the feature vector extracted by the model, and E is the dimension of the output related to our proposal.

**Implementation:** To perform all the experiments, we used the Avalanche framework, which implements the logic to create tasks and evaluate the CL approaches. Regarding the methods, we implemented in Avalanche EmR and Centroids Matching, while the others are already present in the framework. The code containing all the files necessary to replicate the experiments is available here.

## 5.2 Results

**Classification results:** Table 1 summarizes all the results obtained across the experiments, in terms of Accuracy and BWT. We can see that CM significantly improves the results in all the scenarios. The improvements on **TIL** scenarios are significant, by achieving an accuracy that is close to the upper bound set by the cumulative strategy. Moreover, the results are better than all the other methods when it comes to **CIL** scenarios. Surprisingly, the ER approach achieves also good results in all the scenarios, but not as good as the ones achieved by our proposal. The results clearly show the difficulty discrepancy between **TIL** and

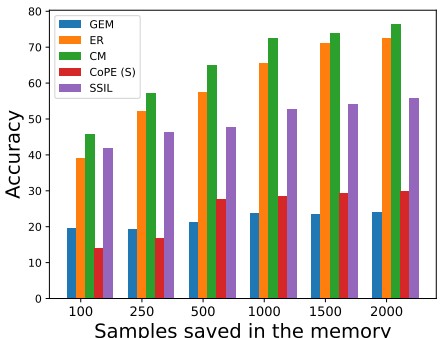

(a) How the accuracy score, obtained on CI-FAR10 **CIL** scenario, changes when the number of the samples saved in the memory changes.

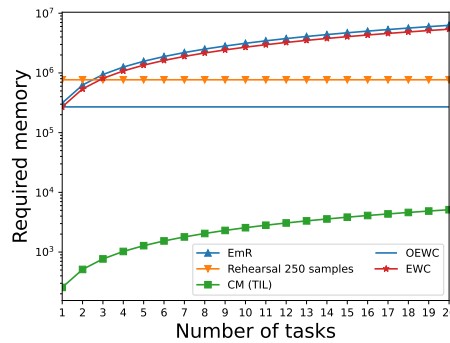

(b) How the required memory grows when the number of tasks increases. The images have size $3 \times 32 \times 32$.

Figure 2: The images show the required memory for each method, as well as how the accuracy changes when the rehearsal memory grows (only for methods that require an external memory containing past samples).

Table 2: The time, in terms of seconds, required to train the models on CIFAR10. The standard deviation as well as the scenario in which the times are obtained are shown.

| Method | Scenario | Task 1 | Task 2 | Task 3 | Task 4 | Task 5 |
|---|---|---|---|---|---|---|
| Naive | **TIL-CIL** | $9.74_{\pm 0.26}$ | $10.84_{\pm 0.27}$ | $10.40_{\pm 0.27}$ | $11.06_{\pm 0.16}$ | $10.50_{\pm 0.33}$ |
| Cumulative | **TIL-CIL** | $11.05_{\pm 0.29}$ | $20.54_{0.28}$ | $30.90_{\pm 1.25}$ | $47.85_{\pm 1.29}$ | $55.98_{\pm 2.32}$ |
| EWC | **TIL-CIL** | $13.61_{\pm 0.020}$ | $17.76_{0.50}$ | $22.32_{\pm 0.28}$ | $28.00_{1.04}$ | $28.89_{\pm 1.55}$ |
| OEWC | **TIL-CIL** | $11.87_{\pm 0.23}$ | $16.73_{\pm 0.18}$ | $16.42_{\pm 0.63}$ | $16.72_{0.61}$ | $17.29_{\pm 0.22}$ |
| ER | **TIL-CIL** | $12.40_{\pm 0.31}$ | $34.00_{\pm 0.72}$ | $61.17_{\pm 1.26}$ | $88.28_{\pm 0.73}$ | $113.78_{\pm 3.62}$ |
| GEM | **TIL-CIL** | $11.79_{\pm 0.48}$ | $26.96_{\pm 0.36}$ | $40.07_{\pm 1.05}$ | $55.58_{\pm 1.11}$ | $70.47_{\pm 0.50}$ |
| EmR | **TIL** | $9.88_{\pm 0.27}$ | $18.97_{\pm 1.85}$ | $43.08_{\pm 0.05}$ | $72.49_{\pm 7.40}$ | $102.20_{\pm 0.35}$ |
| SS-IL | **CIL** | $13.0_{\pm 0.42}$ | $38.60_{\pm 0.70}$ | $47.23_{\pm 2.02}$ | $51.97_{\pm 1.89}$ | $64.93_{\pm 1.93}$ |
| CM | **TIL** | $24.07_{\pm 0.43}$ | $28.62_{\pm 0.17}$ | $35.27_{\pm 0.09}$ | $41.76_{\pm 0.07}$ | $48.90_{\pm 0.43}$ |
| CM | **CIL** | $20.07_{\pm 0.43}$ | $30.62_{\pm 0.17}$ | $45.27_{\pm 0.12}$ | $54.76_{\pm 0.24}$ | $68.08_{\pm 0.27}$ |

**CIL**, because approaches that work well on the first one, drastically fail to overcome the CF in the second one; also, it seems that only methods that use an external memory are capable of achieving good results on **CIL**, and the sole parameters regularization is not enough to fight CF.

**Memory comparison:** the memory required by CL methods is a crucial aspect, and in this section we study how the accuracy score is correlated to this aspect. Figure 2 shows all the aspects related to the memory size. All the results are obtained using the same training configuration used for the main experiments. The image 2a shows the memory usage, correlated with the achieved accuracy score, required by each method when solving CIFAR10 **CIL** scenario. Firstly, we see that GEM is not capable of achieving competitive results even when a large subset of samples is saved, while the others achieve good results even with a smaller memory dimension, and this is probably because a large number of samples is required to correctly estimates the gradients which are used to regularize the training. Regarding the other approaches, we see that our proposal achieves better results even when using a smaller memory. Not all the methods require an external memory containing samples from past tasks, and Image 2b shows how the memory required by all the memory changes when the number of tasks grows. We clearly see that, when it comes to solving **TIL** problems, our proposal requires a smaller memory than all the others. When looking at the results in Table 1 for **CIL** problems, and combining them with the curves in Image 2b, we can conclude that, despite a large amount of memory requested by some methods, few of them are capable of achieving good results; on the other hand, when solving **CIL** scenarios our approach becomes a rehearsal one, and the required memory is almost the same if compared to other rehearsal approaches.

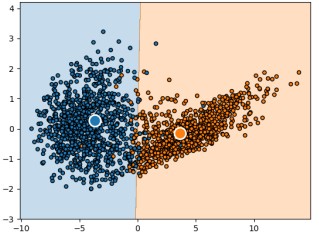
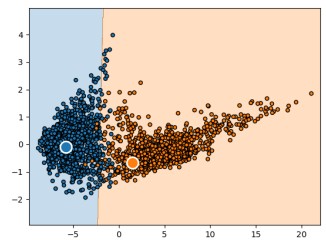
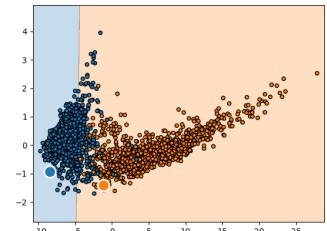

(a) The embedding space when the first task is over.

(b) The embedding space when the third task is over.

(c) The embedding space when the last task is over

Figure 3: The images show how the embeddings spaces, associated with the first task from CIFAR10 **TIL** scenario, change while training on new tasks. We can see that, despite the small changes in the shape of the samples, the overall space is preserved during the whole training. The points are projected into a bi-dimensional space using PCA (Hotelling, 1936). Better viewed in colors.

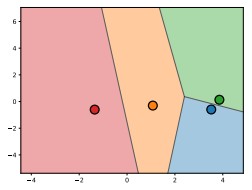
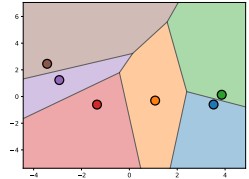
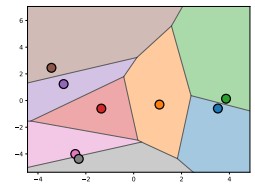
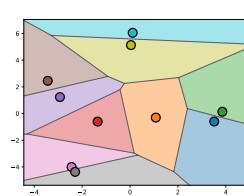

(a) The embedding space when the second task is over.

(b) The embedding space when the third task is over.

(c) The embedding space when the fourth task is over

(d) The embedding space when the last task is over

Figure 4: The images show how the merged embeddings space obtained on CIFAR10 **CIL** changes during the training on all the tasks. The images clearly show that new classes are added without interfering with the ones already present in the space. To visualize clearly the clustering space, we used Voronoi diagrams over the 2D projections of the centroids, obtained using PCA (Hotelling, 1936). The samples are omitted for clarity. Better viewed in colors.

**Training time comparison:** the time required to train a model depends on the Cl approach used to fight the CF. Table 2 contains the time, in seconds, required to train an epoch in each of the 5 tasks of **TIL** CIFAR10 scenario exposed before. We can see that, among the most performing approaches, our proposal is the one that requires the lowest training time: it requires more time than the others during the first task since the extraction of the prototypes from the support set is involved, but then the overall time grows slower than the other approaches. The main difference is the use of the memory in fact when our method is used to solve **CIL** scenarios, the time required is closer to the other rehearsal approaches.

**Analysis of the embedding spaces produced by Centroids Matching:** in this section we analyze how the regularization approach proposed influences the shape of the embedding space produced by a model. In Figure 3 we see how the embedding space, extracted from a model trained on CIFAR10 **TIL** scenario, changes while new tasks are learned: the regularization term is capable of keeping the embedding space almost unchanged, and well separable, during the whole training process. Is also interesting to see how our proposal merges the embedding spaces during the training on a **CIL** scenario, and this aspect is shown in Figure 4. We can see that the classes remain highly separable even in the late stages of the training procedure. The merged space is achieved in an incremental way, by inserting new classes into the existing embedding space, without moving already present centroids. For example, we see that, when passing from the first space to the second one, two new classes are added on the left of the existing embedding space, without interfering with the existing centroids. This is possible because the distance regularization works well, and also because

Table 3: The results, in terms of average and standard deviation calculated over 2 runs, obtained on CIFAR10 **CIL** scenario when varying the merging strategy used, are shown. The results are both in terms of Accuracy and BWT (in the brackets), and both are calculated when training on the last task is over.

| | Task 1 | Task 2 | Task 3 | Task 4 | Task 5 | Accuracy |
|---|---|---|---|---|---|---|
| Scale-Translate | $58.90\,(-33.99)$ | $34.35\,(-50.75)$ | $58.60\,(-26.25)$ | $75.05\,(-20.30)$ | $93.60$ | $64.10\,(-33.99)$ |
| Linear | $43.15\,(-54.70)$ | $51.00\,(-35.40)$ | $46.55\,(-45.80)$ | $63.35\,(-24.95)$ | $89.35$ | $58.68\,(-40.21)$ |
| Offset | $47.25\,(-50.90)$ | $46.30\,(-42.05)$ | $44.35\,(-46.15)$ | $61.25\,(-28.55)$ | $87.85$ | $57.40\,(-41.91)$ |
| None | $43.45(-53.90)$ | $41.70(-46.95)$ | $42.00(-50.06)$ | $65.01(-21.74)$ | $87.80$ | $56.01(-41.70)$ |

the approach is capable of adapting the model to the embedding space, by correctly projecting the centroids and the samples of newer tasks.

### 5.3 Ablation Study

**Comparing merging procedures for CIL scenario.** As exposed in Section 4.2, the merging function used to merge the embedding spaces uses a scale plus transaction function. Here, we study how the choice of the merging function $p_i(\cdot)$ affects the results. To this end, we implemented different functions:

- Scale-Translate: the merging function proposed in Section 4.2.

- Linear: a simple linear layer is used to project the embeddings vector into the new space.

- Offset: an offset is calculated using a linear layer on the embedding, and it is used to shift the embeddings of a given task.

- None: the merging step is performed directly on the embeddings outputted by the model.

For each approach, the weights of the merging networks are shared between the centroids and the embeddings of the same task, to avoid adding many parameters. In Table 3 the results are shown. We see that the Scale-translate approach achieves better results, on average, than all the other approaches, probably due to its inherent capacity to transform the embeddings. The only exception is the second task, in which the approach mentioned above loses more accuracy. Also, as expected, the approach None achieves the worst results but, surprisingly, it is capable of achieving a decent average accuracy.

**How the dimension of the support set affects the training procedure?** Being the support set crucial to our proposal, we expect that its dimension affects the overall training procedure. On the other hand, we also expect that, once an upper bound on the number of support samples is stepped over, the results are not affected anymore, since the same centroids could be calculated using fewer samples. Table 4 shows the results obtained while changing the dimension of the support set. We can clearly see that, under a certain threshold, the results are very close. When the threshold is exceeded, we see a decrease in the achieved accuracy score. This could happen because more images are removed from the training set in order to create the support set, and this negatively affects the results, since some patterns could be missing from the training dataset.

**Different Merging approaches for embeddings and centroids.** In the experiments so far we took into account only the merging approach in which the same merging strategy, and the same weights to perform the merging, are used for both embeddings and centroids. In this section, we also explore the approach in which different strategies are applied separately or when the same strategy uses different weights for centroids and embeddings, in order to understand better which approach is the best one when isolated. The results of this study are exposed in Table 5. We can see that the best results, overall, are achieved when the Scale-Translate merging approach is applied to the embeddings. By combining the results with the ones in the Table 3, we can conclude that the best models are achieved when both the centroids and the embeddings are projected using the same Scale-Translate layer.

**How does $\lambda$ affect forgetting?** In this section we analyze how the parameter $\lambda$, used to balance the regularization term in 5, affects the obtained results. The results are shown in Table 6, and we can see that setting $\lambda$ too high leads the training process to fail when the scenario is a **CIL** one, and inhibits the training

Table 4: The table shows the accuracy, averaged over 2 runs, obtained while changing the number of samples in the support set. The results are calculated using CIFAR10 scenarios; the hyperparameters are the same used in the main experimental section.

| | Support set size | | | | |
|---|---|---|---|---|---|
| | 10 | 50 | 100 | 200 | 500 |
| **TIL** | 90.86 | 90.86 | 91.70 | 89.79 | 90.70 |
| **CIL** | 59.63 | 63.97 | 63.55 | 63.16 | 61.04 |

Table 5: The table shows how combining the merging strategies, used by Centroids Matching, affects the final accuracy obtained on CIFAR10 **CIL**.

| | | Embeddings | | | |
|---|---|---|---|---|---|
| | | S-T | MLP | Offset | None |
| Centroids | S-T | 62.40 | 56.92 | 60.91 | 60.67 |
| | MLP | 62.13 | 56.55 | 59.25 | 62.05 |
| | Offset | 62.84 | 60.94 | 58.37 | 61.82 |
| | None | 61.69 | 62.05 | 60.67 | 59.05 |

Table 6: The results, in terms of average and standard deviation calculated over 2 runs, obtained on CIFAR10 **CIL** scenario when varying the merging strategy used, are shown. The results are both in terms of Accuracy and BWT (in the brackets), and both are calculated when training on the last task is over.

| | 0.01 | 0.1 | 1 | 10 | 100 |
|---|---|---|---|---|---|
| C10 **TIL** | $89.25\,(-6.08)$ | $91.39\,(-8.94)$ | $79.39\,(-10.94)$ | $50.00\,(-12.03)$ | $50.00\,(-12.01)$ |
| C10 **CIL** | $59.75\,(-42.75)$ | $62.32\,(-39.51)$ | $49.57\,(-38.82)$ | $42.19\,(-17.70)$ | $15.95\,(-15.95)$ |

when it comes to **TIL** scenarios (achieving a small forgetting but a small overall accuracy). Moreover, the more the regularization term grows, the more the results degenerate; the same is true also when $\Lambda$ is too small, leading to a model that is not able to remember correctly past tasks. In the end, we can conclude that the best results are obtained when the weight parameter is close and smaller than 0, leading to a model with a balanced trade-off between remembering past tasks and training on the current one.

## 6 Conclusions

In this paper, we proposed an approach to overcome CF in multiple CL scenarios. Operating on the embedding space produced by the models, our approach is capable of effectively regularising the model, leading to a lower CF, requiring no memory when it comes to solving easy CL scenarios. The approach reveals that operating on a lower level, the embedding space, can lead to better CL approaches while having the possibility to analyze the embedding space to understand how the tasks, and classes within, interact.

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
