# OpenReview forum: "Centroids Matching: an efficient Continual Learning approach operating in the embedding space"
_TMLR — Accepted by TMLR_

### Review · Reviewer_NcrE · 2022-08-25

**Summary Of Contributions:**

The authors propose a continual learning method that operates in the Class- and Task-Incremental scenarios by learning through centroid matching and inferring via a nearest neighbour scheme. The authors evaluate their proposal on three commonplace CL classification datasets and carry out several ablation studies to better characterise their design decisions and the behaviour of the proposed approach.

**Requested Changes:**

Summing up from my previous points, I recommend the following revisions (listed by descending importance)
- Expanding and enriching the theoretical comparison with CoPE (De Lange and Tuytelaars, 2021);
- Introducing CoPE as a full-fledged experimental competitor in this paper;
- Introducing better CIL competitors to the experimental section (I recommend choosing among iCaRL, LUCIR, GDumb, LSIL, GSS, DER++, SS-IL, ER-ACE);
- Providing the results for memory-equipped CM in TIL and exploring the trade-off of TIL accuracy/memory occupation linked to removing its memory;
- Mentioning the secondary role of TIL for current literature and highlighting that a relevant portion of contemporary works refer to CIL instead;
- Revising the statements in the related works section concerning regularisation methods and GEM;
- Introducing more citations in support of the claims of the introduction;
- Further explaining/adding experimental evidence in support of the claim in sec. 5.3;
- fixing the typos listed above;
- Reordering tables 2-5.

**Strengths And Weaknesses:**

This work proposes a straightforward and simple approach for continual learning. I found it to be well-written and easy to follow, for which I congratulate the authors. However, I believe that several additional steps need to be taken for the proposal to be clearly outlined and fairly evaluated and for related works to be more accurately represented. I list them in the following:

## Proposal Design
- **Distinction w.r.t. CoPE**: the presented approach is quite similar to CoPE, proposed in (De Lange and Tuytelaars, 2021), as also mentioned in the text. Both approaches follow the same idea and basically optimise the same loss, but I reckon there are some differences (i.e.: CoPE proposing a specific heuristic for centroid update vs. CM introducing an architectural solution by having a separate $f_i$ for each task). I strongly recommend expanding the discussion of (De Lange and Tuytelaars, 2021) to let the reader better understand how these two works differ. This needs to be complemented with a thorough experimental analysis, thus also providing a quantitative counterpart to the analysis (more on that below).
- **Usage of replay memory**: I find it somewhat convoluted that the authors propose two separate methods for the TIL and CIL settings. In my understanding, the only rationale for this dichotomy is that TIL is significantly easier than CIL, thus not requiring the introduction of a separate memory buffer. Nonetheless, I expect the memory-equipped version of CM to outperform memoryless CM in TIL too. Instead of proposing two practically identical methods, I believe that a clearer exposition could be made by only proposing memory-equipped CM and then investigating the possibility of removing the memory in TIL in a separate ablation study.

## Fair Evaluation
- **Better competitors are needed**: the selection of evaluated competitors in section 5.1 is not reflective of the current state of the art in continual learning classification. The only competitor which is not designed for the TIL setting is ER (still, its effectiveness varies considerably depending on implementation details [1, 2]). In a modern paper targeting the CIL setting, I expect the inclusion of several of the following strong recent baseline methods: iCaRL [3], LUCIR [4], GDumb [5], LSIL [6], GSS [7], DER++ [8], SS-IL [9], ER-ACE [10]. As discussed above, I believe that the experimental inclusion of CoPE (De Lange and Tuytelaars, 2021) is also of the utmost importance.
- **Number of epochs**: I could not find any information about the number of training epochs, which is a pretty important factor conditioning the results that are obtained in CL classification. This information should be featured clearly in the text.
- **Uneven memory in comparisons**: from my understanding, the authors write that the following memory buffers are used:
	- CIFAR10: GEM uses 500 examples per task (so, 2500 in total?), EmR uses 200 (1000 in total?), ER uses 1000 (in total? per task? this is not clear, CM uses 500 (also, unclear whether this is the total or per-task number).
	- CIFAR100: GEM uses 1000 examples per task (10000 in total, 1/6 of the overall dataset?), EmR uses 200 (1000 in total?), ER uses 500 (in total? per task?), CM uses 500 (in total? Per task?).
	- TinyImageNet: GEM uses 1000 examples per task (10000 in total, 1/6 of the overall dataset?), EmR uses 200 (1000 in total?), ER uses 1000 (in total? per task?), CM uses 500 (in total? Per task?).

I think that this pieces of information should be provided more clearly; from my understanding, I see a remarkable disparity in the allowance of replay buffer across these methods, which makes the proposed experimental comparison hard to understand. I would recommend using at least the same memory size for all competitors.
- **Memory usage**: in line with and beyond the previous point, I agree with what is stated in Sec. 5.1, namely that different competitors employ widely varying working memory for their training and operation. I believe that it would be very useful for the reader if a memory occupation/performance analysis graph were provided for a couple of experimental CIL settings. Such a comparison clearly allows for a direct comparison across all methods. It would also be interesting to highlight the support set memory footpring (which is not needed by the competitors) in this comparison.
- **Batch normalisation trick**: is the batch norm trick described in the first paragraph of page 8 also employed for other competitors in TIL benchmarks? If not, I would be very interested in seeing how it affects the performance of e.g. ER, given that the batch norm drift is factor known to affect CL results in a major fashion [11].

## Exposition of related works
- **Improving the characterisation of TIL vs CIL**: the authors suggest that mostly of the experimental evaluation in CL happens in the TIL scenario and "fewer methods have been proposed to efficiently solve [CIL]". I believe that this is no longer reflective of the current state of literature, where CIL is widely understood - as also stated by the authors - to be much more interesting than TIL [12, 7, 13, 14] and a consistent number of baselines targets this setting directly [3, 4, 5, 6, 7, 8, 10]. I would argue that in the last two/three years it is much easier find papers referring to CIL in major conferences than TIL. I would suggest accounting for this fact in the main text and de-emphasising the role of TIL.
- **Improving the characterisation of regularisation and rehearsal methods**: in the related works section, I do not agree with some proposed characterisation of regularisation and rehearsal methods. For the former, it is written that they "fight CF by forcing the model's parameters [...] to be as close as possible to the [previous optimum]". This description is not necessarily fitting to methods that regularise the output instead of parameters, such as LwF. Similarly, I do not at all agree with the statement that "the most representative method in [rehearsal methods] is [GEM]". In fact, I would rather characterise it as a somewhat hybrid solution that uses replay examples to compute a regularisation/projection term instead of simply replaying stored data. Furthermore, I do not understand the statement that "[GEM] was the first method that can improve the scores obtained on past tasks": this is not true, as ER (which was originally proposed in the 90s) also fits this description, if one performs few epochs on a given task, then the model is likely not to perfectly fit the data and thus to improve its performance on it upon replay. I recommend adjusting these sentences to better characterise regularisation and rehearsal methods.
- **Introduction could use more citations**: as a minor suggestion, I believe that the introduction presents multiple claims which are not supported through citations in literature and thus end up appearing as quite unjustified. I suggest introducing further citations so as to make it possible for the reader to go in depth on specific aspects of CL by themselves. Examples of sections that would need it are "we can't save all past samples encountered during training", "class-incremental [...] is much harder to deal with, being closer to a real-world scenario".

## Misc Minor Points
- **Unclear claim in sec. 5.3**: I do not entirely understand the claim made in the first paragraph of sec. 5.3, which states that there exists a critical number of samples beyond which results don't improve since the same centroids are computed with fewer samples. I am not sure that this is immediately evident, could it be experimentally verified that one obtains the same centroids for different sizes of the support set?
- **Reordering ablation tables**: I would suggest presenting the Tables 2-5 in the same order they are cited in text.
- **Typos**: while reading, I found the following typos:
	- the spelling of Matthias De Lange's name is inconsistent in citations;
	- sec. 4.1, two lines above eq. 1: centroids -> centroid
	- sec. 4.1, one line below eq. 5: the final centroids, for the classes in the task, are calculated -> the final centroids for the classes in the task are calculated
	- sec. 4.2, in eq.7 $Y_i$ seems to have a different meaning w.r.t. $Y_i$ in eq.2, I would recommend choosing a different symbol
	- sec. 4.2, in eq.7 it seems that the second operand of d is $p_i(x)$, should it not be $p_i(e_i(x))$? Is it operating on an embedding or on the input?
	- sec. 4.2, one line above eq.10: corresponding class -> corresponding task
	- why is there no variance in the results of EWC and OWC on TinyImageNet?

[1] Chaudhry, Arslan, et al. "On tiny episodic memories in continual learning." ICML workshop 2019.
[2] Buzzega, Pietro, et al. "Rethinking experience replay: a bag of tricks for continual learning." 2020 25th International Conference on Pattern Recognition (ICPR). IEEE, 2021.
[3] Rebuffi, Sylvestre-Alvise, et al. "icarl: Incremental classifier and representation learning." Proceedings of the IEEE conference on Computer Vision and Pattern Recognition. 2017.
[4] Hou, Saihui, et al. "Learning a unified classifier incrementally via rebalancing." Proceedings of the IEEE/CVF Conference on Computer Vision and Pattern Recognition. 2019.
[5] Prabhu, Ameya, Philip HS Torr, and Puneet K. Dokania. "Gdumb: A simple approach that questions our progress in continual learning." European conference on computer vision. Springer, Cham, 2020.
[6] Wu, Yue, et al. "Large scale incremental learning." Proceedings of the IEEE/CVF Conference on Computer Vision and Pattern Recognition. 2019.
[7] Aljundi, Rahaf, et al. "Gradient based sample selection for online continual learning." Advances in neural information processing systems 32 (2019).
[8] Buzzega, Pietro, et al. "Dark experience for general continual learning: a strong, simple baseline." Advances in neural information processing systems 33 (2020): 15920-15930.
[9] Ahn, Hongjoon, et al. "Ss-il: Separated softmax for incremental learning." Proceedings of the IEEE/CVF International Conference on Computer Vision. 2021.
[10] Caccia, Lucas, et al. "New insights on reducing abrupt representation change in online continual learning." ICLR 2022.
[11] Pham, Quang, Chenghao Liu, and Steven Hoi. "Continual normalization: Rethinking batch normalization for online continual learning." ICLR 2022.
[12] Farquhar, Sebastian, and Yarin Gal. "Towards robust evaluations of continual learning." ICML workshop 2018.
[13] Masana, Marc, et al. "Class-incremental learning: survey and performance evaluation on image classification." arXiv preprint arXiv:2010.15277 (2020).
[14] Belouadah, Eden, Adrian Popescu, and Ioannis Kanellos. "A comprehensive study of class incremental learning algorithms for visual tasks." Neural Networks 135 (2021): 38-54.

---

> ### Author Response · Authors · 2022-09-10
> **Answer to Reviewer NcrE**
>
> We would like to thank the reviewer for the in deep revision. We addressed all the points raised in the just uploaded manuscript. Moreover, we answer some points here:
>
> > Usage of replay memory: I find it somewhat convoluted that the authors propose two separate methods for the TIL and CIL settings. In my understanding, the only rationale for this dichotomy is that TIL is significantly easier than CIL, thus not requiring the introduction of a separate memory buffer. Nonetheless, I expect the memory-equipped version of CM to outperform memoryless CM in TIL too. Instead of proposing two practically identical methods, I believe that a clearer exposition could be made by only proposing memory-equipped CM and then investigating the possibility of removing the memory in TIL in a separate ablation study.
>
> In our preliminary experiments, we observed that a small external memory is not enough to regularize in a TIL scenario. This happens because the Prototypical Networks have been proposed to perform, mainly, few-shot learning, and they are capable of rapidly changing the weights to adapt to new samples.  On the other hand, the model requires external memory to solve CIL tasks, otherwise, the score on past tasks could drop rapidly to zero. For this reason, the two approaches are separate.
>
> > Unclear claim in sec. 5.3: I do not entirely understand the claim made in the first paragraph of sec. 5.3, which states that there exists a critical number of samples beyond which results don't improve since the same centroids are computed with fewer samples. I am not sure that this is immediately evident, could it be experimentally verified that one obtains the same centroids for different sizes of the support set?
>
> The main point is that we don’t want the model to produce the same centroids regardless of the support set, but we need, in theory, only a subset of samples to produce meaningful centroids used for training. Table 4 contains the results associated with that claim.

---

> > ### Comment · Reviewer_NcrE · 2022-09-28
> > **Further Clarification**
> >
> > I thank the authors for taking the time to address some of the major points I raised in my review. I welcome the addition of COPE and SS-IL, to competitors, which somewhat improve the presentation of the overall experimental section.
> >
> > However, I see COPE results are very low in Tab.1, which I believe is counter-intuitive. To produce a final evaluation, I need to have further clarifications on this point: I do not clearly understand whether the authors ran COPE experiments in the single epoch settings, while other methods have more than one epoch at their disposal. If this were the case, the evaluation would clearly not be fair. Alternatively, it is possible that COPE has a different case to underperform in spite of its similarity to CM, but I strongly feel that a hypothesis for it should be at least be offered in the comment.

---

> > > ### Author Response · Authors · 2022-09-28
> > > **Clarification about CoPE results**
> > >
> > > Since we considered this point at length when preparing the review, we provide a brief rationale of our point of view below.
> > >
> > > First, we note that CoPE was proposed for a different setup (what the authors call "data incremental learning", DIL), which is more general that TIL and CIL since the task identifier is not provided at training time. However, the experimental evaluation and the official code are provided only for a fully streaming setup. To this end, we decided that the most fair way of comparing CoPE with our model was following as closely as possible the original evaluation of the CoPE paper, instead of testing novel variants (e.g., by building a data stream which is composed of multiple copies of each task, each corresponding to a single "epoch" in our algorithm). Since CoPE is designed for a harder situation, in fact, it is reasonable that his results are lower.
> > >
> > > Regarding the results, we note that the ones proposed in our paper are aligned with the ones in the original CoPE paper when the comparison is feasible. Comparing the results obtained on CIFAR10, we see that we achieve 49% (Table 1) and in the original CoPE paper, the authors achieve around 55% (Image 3b), using the same memory size (the gap between the scores can arise from weights initialization or the task creation process, or any other step that contains a degree of randomness). Also, the results are very similar when the dimension of the memory is changed (Image 2a in our paper and  Image 3b in the CoPE paper). The problem arises on CIFAR100, but we underline that the splitting approach of the two papers differ: we create 10 tasks with 10 classes in each one, while in CoPE the authors used the dataset to build 20 tasks, each one with 5 classes. These different splits lead to different problems, and the ones used in our paper are more difficult since each batch contains fewer images for a single class. Regarding TinyImageNet, we have no results to compare since it is not used in CoPE.
> > >
> > > CoPE has a parameter called "iterations", that governs how many times each batch, when observed, is used to update the model and the prototypes, but as the authors noted in the original paper this can lead to worse results due to overfitting, and in the original paper this value is set to one (the batch is used to train the parameters and then discarded).
> > >
> > > In the end, we think that the results are correct in the sense outlined above and aligned with the CoPE paper, which has been implemented using the original code made available by the authors. We would be happy to replicate additional experiments under other setups if the reviewer requests them.

---

> > > > ### Comment · Reviewer_NcrE · 2022-10-02
> > > > **Response**
> > > >
> > > > Thank you for taking the time to detail this aspect of the evaluation.
> > > > I believe that the disparity in the epoch regime (and - to a lesser extent - in the number of tasks) chosen for CM w.r.t. CoPE does not allow for a clear comparison between the two methods. I would recommend either trying and applying CoPE in the same setting as CM (ie. 10 epochs-per-task and 10 tasks for CIFAR-100) or removing CoPE from Table 1 and separately evaluate CM in the same setting as CoPE.

---

> > > > > ### Author Response · Authors · 2022-10-04
> > > > > **CoPE results on CIL scenario**
> > > > >
> > > > > In order to answer, we performed an additional test for CoPE in the same scenario (CIL) as the other methods (same number of epochs, same split), and we modified the paper accordingly. We observe that the results are slightly worse than the ones achieved on a stream setting, and we hypothesize this could happen due to two different reasons: 1) the epochs make the prototypes less general because are constantly trained on past data and for more iterations, 2) the regularization approach, implemented as two loss functions in the paper, is not strong enough when it comes to the CIL scenario.

---

> > > > > > ### Comment · Reviewer_NcrE · 2022-10-05
> > > > > > **Thanks**
> > > > > >
> > > > > > Thank you for the additional results. I am now satisfied with the provided results and will recommend acceptance.

---

### Review · Reviewer_7Yto · 2022-08-26

**Summary Of Contributions:**

The paper studies task incremental and class incremental continual learning where the authors propose to learn and use class centroids for classification. The Centroid Matching (CM) framework uses a held-out set to calculate the centroids during the training and then uses(distribution of) the euclidean distance of the inputs and centroids as the objective. In addition, in the task incremental setting, a regularization term will be used that penalizes new task data for being close to previous tasks' centroids. In contrast, in the class incremental setting, since the classification head is shared, CM uses an episodic memory and projects embeddings into a shared space. The authors then validate the effectiveness of the CM approach in both task and class incremental setup, in addition to some ablation studies.

**Requested Changes:**


I believe the following changes are crucial (see above for details):
1- Including more recent/related baselines.
2- Providing runtime comparison between different methods.
3- Explain the different memory sizes or use a similar memory size for all methods.

**Strengths And Weaknesses:**






## Strengths

1. The paper does a great job of explaining the method very clearly.

2. The proposed method performs well under task and class incremental setups.



## Weaknesses

1. **Baselines**: My first concern is about the baselines. I believe there have been several methods proposed in the past two years that perform significantly better than the classic CL methods, such as EWC and GEM. In addition, while CoPE is motivated by streaming settings, I believe a comparison would be nice since the fundamental assumptions in both works are similar.

2. **Computation**: From the paper, it seems like to calculate the regularization term (Eq. (4)), we need both the previous backbone and heads. So I think the runtime is twice as the naive model. However, there are no runtime comparisons between the baselines, and I think it is necessary to compare the runtimes/computation requirements.

3. **Memory Size**: In Sec. 5.1, the authors mention they use different memory sizes for different methods. For instance, for the CIFAR-100 benchmark, GEM uses 1000 examples, ER uses 500 examples, and EmR uses 2000 examples. I think this makes the evaluation confusing. Could the authors explain the rationale behind this choice?

---

> ### Author Response · Authors · 2022-09-10
> **Answer to  Reviewer 7Yto**
>
> We would like to thank the reviewer for the review. We addressed all the weak points in the uploaded manuscript.
> In particular:
>
> >  Including more recent/related baselines.
>
> To this end, we added two more baselines: SS-IL [1] and CoPE [2]. The first one operates only under CIL scenarios, while the latter was proposed to alleviate CF in a stream of data, but can also be used to solve CIL scenarios (as in the paper)
>
> > Providing run-time comparison between different methods.
>
> We added a new table, with the approximated run-time required by each model
>
> > Explain the different memory sizes or use a similar memory size for all methods.
>
> When performing the main experiments, we used different dimensions because each approach differs from the others. To do so, we followed the associated papers. Moreover, we also have an image in which the methods are compared when the memory budget is the same. We made it more clear in the text.
>
> [1] Ahn, Hongjoon, et al. "Ss-il: Separated softmax for incremental learning." Proceedings of the IEEE/CVF International Conference on Computer Vision. 2021.
> [2] De Lange, Matthias, and Tinne Tuytelaars. "Continual prototype evolution: Learning online from non-stationary data streams." Proceedings of the IEEE/CVF International Conference on Computer Vision. 2021.

---

### Review · Reviewer_uPg7 · 2022-08-29

**Summary Of Contributions:**

**Problem**:
This work addresses the problem of continual learning *i.e.* learning a sequence of tasks without forgetting/deteriorating performance on the previous. The work focuses on the standard setting of a sequence of image classification tasks.

**Solution**: The paper proposes a solution called "centroid matching" which involves learning and memorizing the centroids of the classes for each task. In addition to this, the training involves novel regularization methods which help in alleviating issues of forgetting. The regularization ensures that previously learned centroids are not perturbed while training on novel tasks.

**Broader Impact Concerns:**

The paper does not include a broader impact section. However, I do not see any obvious ethical concerns in this work.

**Requested Changes:**

Some suggestions for improving the paper:
#### From weaknesses in Approach above
- Analysis of sensitivity to the order of tasks might be interesting to add. For example, if the subsets of cifar100 were permuted, does the final overall accuracy stay the same?
- Analysis of sensitivity to more significant changes in distribution might be interesting to compare with existing works. For example, training on a subset of cifar10 and then a subset on tinyimagenet and then a subset of cifar100.

#### From weaknesses in Results above
- Could you add an explanation or experimental analysis for the worse BWT?
- Is there a reason that De Lange and Tuytelaars (2021)  cannot be included in the comparisons?



**Strengths And Weaknesses:**

# Writing, Problem Statement, Motivation, Related Work
### Strengths
The paper does a thorough job of presenting the problem statement in a concise manner. The introduction explains the continual learning problem and issues of catastrophic forgetting. This work extends beyond the standard "Task Incremental Learning" version of continual learning (known task boundaries and identities) to the "Class Incremental Learning" formulation (unknown task identities for samples). While the text proposes two somewhat different approaches for the two settings, the text clearly presents the distinction while limiting any confusion.

### Weaknesses
- Portions of the text are redundant and repetitive. For example, the distinction between task incremental and class incremental learning appears in Sec 1, Sec 2, Sec 3 and Sec 4.2.
- While most relevant literature has been cited, I think the discussion on the distinctions between existing work and the proposed work can be elaborated further. For example, Prototypical Networks serves as an important backbone for this work and has been just cited in passing. There also seem to be a lot of similarities to  De Lange and Tuytelaars (2021). This work also has been very briefly mentioned.




# Approach

### Strengths, Novelty
The presented approach is very intuitive to follow. The key component is a novel regularization used in conjunction with Prototypical Networks trained in a continual learning setting. The regularization ensures that the previously learned prototypes (or centroids) are unperturbed. This overall idea is simple, novel and might be of general interest.

The proposed approach is also fairly general and can be applied in both the continual learning problem formulations - Task Incremental Learning and Class Incremental Learning. Due to the difference in the two settings the paper proposes slightly modified implementations of the regularization. I believe these ideas provide interesting and useful guidelines for implementing general regularization methods in the continual learning framework.

Unlike some previous work on continual learning, the proposed approach also makes efficient use of memory during training.

### Weaknesses, Concerns, Questions
- The objective in Eq 4 relies on images from novel tasks for ensuring that the embedding of previously seen images is unperturbed. However, this might not be very effective if the images are from significantly different distributions.
- Since the amount of regularization increases with number of additional tasks, it might be the case that the model performs poorly on the later tasks due to over-regularization. This might also mean that the training is sensitive to the order of tasks seen.



# Results

### Strengths

The proposed work outperform the accuracy of reported existing works on numerous continual learning benchmarks (CIFAR10, CIFAR100, TinyImageNet). Furthermore, the reported accuracy is higher for both formulations of the problem (TIL, CIL).

The paper presents a thorough analysis of the different features of the proposed work. The analysis of memory consumption shows that the performance of the proposed work scales better compared to existing work. The memory consumption of the proposed work also increases slower with increasing number of tasks.

The results present very interesting visualizations of the learned feature embeddings. The visualizations show that the centroids are indeed unperturbed when training on novel tasks. It also demonstrates that the model learns to cleverly assign novel centroids to minimize conflicts with previously learned centroids.

### Weaknesses
- The BWT metric seems to be a better metric for measuring the amount of forgetting. However, the proposed work seems to underperform on this metric while being better on overall accuracy.
- To the best of my understanding, De Lange and Tuytelaars (2021) is the closest related work. However, it looks like experimental comparisons to this work has been omitted.

---

> ### Author Response · Authors · 2022-09-10
> **Answer to Reviewer uPg7**
>
> We would like to thank the reviewer for the review. We addressed all the weak points in the uploaded manuscript.
>
> Also, we answer some raised weak points below:
>
> > While most relevant literature has been cited, I think the discussion on the distinctions between existing work and the proposed work can be elaborated further. For example, Prototypical Networks serve as an important backbone for this work and have been just cited in passing. There also seem to be a lot of similarities to De Lange and Tuytelaars (2021). This work also has been very briefly mentioned.
>
> We added a discussion on how the approaches are similar to our proposal
>
> > Since the amount of regularization increases with number of additional tasks, it might be the case that the model performs poorly on the later tasks due to over-regularization. This might also mean that the training is sensitive to the order of tasks seen.
>
> This is indeed true, and it is true for almost all the CM approaches since the capacity of the neural models is limited. Regarding the order of the classes, we build the scenarios after shuffling the order of the classes, in order to take into account also the raised problem.
>
> > Analysis of sensitivity to the order of tasks might be interesting to add. For example, if the subsets of cifar100 were permuted, does the final overall accuracy stay the same?
>
> As stated in the text, we shuffled the classes before creating the scenarios. In this way, the resulting CL scenario will result in multiple shuffled tasks, leading to a new scenario in each experiment. However, we think that a study on how the shuffling affects the methods is outside the scope of our paper.
>
> > Analysis of sensitivity to more significant changes in distribution might be interesting to compare with existing works. For example, training on a subset of cifar10 and then a subset on tinyimagenet and then a subset of cifar100.
>
> Despite being an interesting research direction, we have no time to perform all the experiments before the revision due date. We leave this point for future work.
>
> > Could you add an explanation or experimental analysis for the worse BWT?
>
> The BWT is important to evaluate a CL approach, but cannot be evaluated alone, because a method can achieve low BWT but also low Accuracy, meaning that the approach constraints the training too much, leaving no space for improvement when training on new tasks. We made it more clear in the text.
>
> > Is there a reason that De Lange and Tuytelaars (2021) cannot be included in the comparisons?
>
> The approach was omitted because it operates in a different scenario (a stream of batches, instead of tasks). However, since in the original paper the approach was compared also to methods operating in the CIL scenario, we included it in our paper, following the same approach used in the original one.

---

### Decision · Action_Editors · 2022-10-09

**Recommendation:** Accept with minor revision

**Comment:**

This paper proposes a method for mitigate catastrophic forgetting in task-incremental and class-incremental learning settings. The approach builds upon Prototypical Networks, which performs classification using a softmax over distances to prototypes formed by taking the centroid of the embeddings of examples belonging to each class. In the task-incremental learning setting, each task gets a new head, and the model is regularized so that the embeddings of the new examples using the old heads match the embeddings obtained using a frozen clone of the model made before initiating training on the new task. In the class-incremental learning setting, all tasks are additionally projected to a common embedding space where all classes can be discriminated, and a small set of examples from previous tasks are used to ensure that it is possible to separate the classes in the common embedding space.

Reviewers noted that, although the proposed approach is similar to CoPE (De Lange and Tuytelaars, 2021), it is novel and performs well, but requested additional baselines. The authors have added these baselines, and all reviewers now recommend acceptance.

I am satisfied with the revisions the authors have made over the course of the review process and recommend acceptance. There are a few changes that I believe could further strengthen the paper:

- Given the discussion between the authors and Reviewer NcrE of what constitutes a fair CoPE baseline, I feel that the paper should describe the CoPE baseline setup in greater detail than it currently does, and include both the single-epoch and multi-epoch CoPE results that they provided over the course of the response period.
- I believe there is a missing log in Eq. 9. Furthermore, the first paragraph of Section 4.2 implies that the loss on the shared embedding space in the CIL setting is applied in addition to the other losses applied in the TIL setting, but these losses do not appear in Eq. 9. It would be helpful to clarify these points.
- As Reviewer NcrE privately in their recommendation, there is a red CoPE label in the legend of Figure 2a, but no red bars in the graph.

**Audience:**

Yes. The proposed approach is methodologically novel, simple, and works well.

**Claims And Evidence:**

Yes, the claims are well-supported. In their initial reviews, the reviewers raised some concerns regarding the adequacy of the baselines. The authors convincingly addressed these points in their responses.